# Dynamic Changes in Circulating Tumor DNA During Immunotherapy for Head and Neck Cancer: SHIZUKU-HN Study

**DOI:** 10.3390/ijms26010235

**Published:** 2024-12-30

**Authors:** Rika Noji, Kohki Tohyama, Shin Nakamura, Takahiro Naito, Yu Oikawa, Takeshi Kuroshima, Hirofumi Tomioka, Yasuyuki Michi, Sadakatsu Ikeda, Takahiro Asakage, Masahiko Miura, Yasuo Hamamoto, Hiroyuki Harada, Yoshihito Kano

**Affiliations:** 1Department of Oral and Maxillofacial Surgical Oncology, Division of Health Science, Graduate School of Medical and Dental Sciences, Institute of Science Tokyo, 1-5-45 Yushima, Bunkyo-Ku, Tokyo 113-8510, Japan; nojiri1207.osur@tmd.ac.jp (R.N.);; 2Department of Medical Oncology, Institute of Science Tokyo, 1-5-45 Yushima, Bunkyo-Ku, Tokyo 113-8510, Japan; 3Department of Dental Radiology and Radiation Oncology, Graduate School of Medical and Dental Sciences, Institute of Science Tokyo, 1-5-45 Yushima, Bunkyo-Ku, Tokyo 113-8510, Japan; 4Department of Precision Cancer Medicine, Center for Innovative Cancer Treatment, Institute of Science Tokyo, 1-5-45 Yushima, Bunkyo-Ku, Tokyo 113-8510, Japan; 5Department of Head and Neck Surgery, Institute of Science Tokyo 1-5-45 Yushima, Bunkyo-Ku, Tokyo 113-8510, Japan

**Keywords:** head and neck squamous cell carcinoma (HNSCC), liquid biopsy, circulating tumor DNA (ctDNA), immune checkpoint inhibitor (ICI), variant allele frequency (VAF), dynamic monitoring, Center for Cancer Genomics and Advanced Therapeutics (C-CAT)

## Abstract

Immune checkpoint inhibitors (ICIs) are effective in treating recurrent/metastatic head and neck squamous cell carcinoma (HNSCC), but only 20% of patients achieve durable responses. This study evaluated circulating tumor DNA (ctDNA) as a real-time biomarker for monitoring treatment response in HNSCC. The SHIZUKU-HN study prospectively collected and analyzed serial plasma samples (n = 27) from HNSCC patients undergoing ICIs, using Guardant360 to assess ctDNA variant allele frequency (VAF) and genetic mutations. Tumor volumes were quantified using 3D reconstruction of CT scans, and data from Japan’s C-CAT database (n = 2255) provided insights into ctDNA testing in HNSCC. C-CAT data showed that ctDNA testing was underutilized, performed in only 7% of head and neck cancer cases. In SHIZUKU-HN, mean VAF significantly correlated with tumor volume (Spearman’s ρ = 0.70, *p* = 0.001), often preceding radiographic progression. BRAF and APC mutations disappeared in partial responders, while GNAS mutations varied. EGFR and PIK3CA amplifications, detectable via ctDNA but missed in tissue biopsies, indicated emerging resistance mechanisms. The SHIZUKU-HN study demonstrates the potential of ctDNA as a dynamic biomarker in HNSCC, offering early insights into treatment efficacy and informing personalized ICI therapy.

## 1. Introduction

Immune checkpoint inhibitors (ICIs) have revolutionized the treatment landscape for recurrent/metastatic head and neck squamous cell carcinoma (R/M HNSCC), establishing themselves as a standard therapeutic option. However, only about 20% of patients achieve a durable clinical response to ICIs, underscoring the urgent need to identify which patients are most likely to benefit from these therapies [1,2]. Predictive biomarkers like programmed death-ligand 1 (PD-L1) expression and tumor mutation burden (TMB) have shown inconsistent predictive value [3,4,5], highlighting the limitations of current methods in guiding ICI therapy.

Although comprehensive genomic profiling (CGP) of tissue samples offers insights into genetic alterations, it fails to capture the dynamic nature of tumor evolution, particularly under therapeutic pressure. Liquid biopsy, or analyzing circulating tumor DNA (ctDNA) from blood samples, has emerged as a promising non-invasive approach that allows real-time monitoring of tumor dynamics [6,7]. ctDNA reflects cancer-specific genetic alterations, providing a snapshot of the tumor’s genomic landscape that may change rapidly during treatment [8]. In Japan, blood-based CGP tests such as FoundationOne^®^ Liquid and Guardant360 have gained regulatory approval, expanding the clinical utility of ctDNA for cancer management. The ctDNA dynamics during systemic treatment may be prognostic for clinical outcomes, even for treatments not associated with specific actionable genomic alterations, such as ICI [9] or broad-spectrum small molecule inhibitors [10]. Despite the expanding application of liquid biopsy in other cancer types, such as non-small cell lung cancer and bladder cancer, its potential clinical utility in HNSCC remains underexplored, largely because the use of therapies targeted against genomic alterations has been limited. Currently, no standardized blood biomarkers are available for routine assessment of therapeutic efficacy in HNSCC patients. Addressing this gap, the SHIZUKU-HN study aims to evaluate the potential of ctDNA as a real-time biomarker for monitoring treatment response in HNSCC patients undergoing ICI therapy, using data from Japan’s national genome database established by the Cancer Center for Advanced Cancer Genome Therapy (C-CAT) [11,12].

The SHIZUKU-HN study prospectively monitors the dynamic changes in key mutations, including BRAF, APC, GNAS, EGFR, and PIK3CA, which are implicated in treatment response and resistance. By examining the correlation between ctDNA variant allele frequency (VAF) and total body tumor volume, we hypothesize that ctDNA could serve as a dynamic marker of tumor burden and therapeutic efficacy. This study provides novel insights into the use of liquid biopsy for personalized monitoring in HNSCC, aiming to refine patient selection for ICIs and improve clinical outcomes.

## 2. Results

### 2.1. Utilization Patterns and Genetic Alterations in ctDNA from the C-CAT Database

From June 2019 to December 2023, among the 65,410 patients registered in the C-CAT, 2255 were diagnosed with head and neck cancer (Figure 1a,b). The use of panel tests (F1LCDx) with blood specimens was lower in head and neck cancer (7%) compared to other cancers, such as bowel cancer (9%), pancreatic cancer (26%), biliary tract cancer (20%), and breast cancer (16%).

In HNSCC, frequently mutated genes detected in plasma samples included TP53 (65.7%), ATM (43.3%), DNMT3A (34.4%), CHEK2 (23.9%), PIK3CA (22.4%), ASXL1 (22.4%), CDKN2A (16.5%), and GNAS (16.4%) (Figure 1c). Mutations in DNMT3A, CHEK2, and ASXL1—often associated with clonal hematopoiesis of indeterminate potential (CHIP)—were common, reflecting the potential influence of aging-related phenomena on ctDNA results [13,14]. Most plasma mutations matched those observed in tissue samples, including frequently mutated genes such as TP53, TERT, CDKN2A/B, and Notch1 in HNSCC. However, amplifications of CCND1, PIK3CA, and EGFR, as well as deletions of CDKN2A/B, all of which were observed in tissue samples, were not detected in plasma.

### 2.2. Correlation Between ctDNA VAF and Tumor Volume in the SHIZUKU-HN Study

A total of 27 serial plasma samples from 10 patients were analyzed. Three patients discontinued the study after the second time point due to rapid disease progression. The median follow-up period from the first ctDNA sample collection was 65 days (range: 28–273 days), with radiographic evaluations conducted within 0–1.6 months of each ctDNA collection. Pathogenic alterations were detected in 8 out of the 10 patients. Two patients had no detectable VAF of pathogenic variants at any treatment time points. One patient, who exhibited SD with a small target lesion in the cervical lymph node (<1 cm^3^), only had variants of unknown significance (VUS). The other patient, despite having a substantial target lesion (31.3 cm^3^ in the lung), also showed no detectable mutations, eventually achieving PR, indicating potential low ctDNA sensitivity from lung lesions.

We analyzed whether the VAF of ctDNA correlates with tumor volume. We defined pVAF as the VAF value of the pathogenic gene and aVAF as the VAF value of all detected gene mutations, including VUS and synonymous mutations. Of the mean, maximum, and total values of pVAF or aVAF, mean pVAF demonstrated the strongest correlation with tumor volume (Spearman’s ρ = 0.667, *p* = 0.001; Figure 2a), outperforming aVAF in predictive accuracy.

### 2.3. Correlation Between ctDNA VAF and Treatment Response

The best clinical responses observed were PR in three patients (30%), SD in one (10%), and PD in six (60%). In three of all PD patients, an early increase in VAF for all detected variants, including TP53, was observed from the second time point, preceding clinical signs of progression. This early rise in VAF showed disease progression before traditional imaging methods (Figure 2b). This finding supports the hypothesis that changes in VAF of ctDNA are associated with total body tumor volume. The patterns of mean pVAF differed by response type: PR and SD patients exhibited stable VAF levels and were associated with disease control. In contrast, PD patients often showed elevated VAF levels early in progression (Figure 2c).

### 2.4. Dynamic Changes in BRAF and APC Mutations During ICI Treatment

The most frequently detected mutated gene was TP53 (44%), followed by the TERT promoter (8%), PIK3CA (8%), GNAS (4%), and CDKN2A (4%) (Figure 3a). Notably, no CHIP-associated mutations were detected, unlike in the broader C-CAT data.

During ICI treatment in PR patients, BRAF and APC mutations shifted as the tumor responded to or resisted treatment. In patient HN01, BRAF mutations were detected at the second evaluation, consistent with PR. However, as the patient progressed to PD, the BRAF mutation disappeared by the third evaluation time point. In patient HN12, an inactivating APC mutation present at baseline disappeared during treatment, coinciding with sustained PR (Figure 3b).

### 2.5. GNAS Mutations and Their Controversial Role in Treatment Response

GNAS mutations presented a complex scenario, appearing in both PR and PD cases with differing implications. In HN12, GNAS mutations emerged despite ongoing PR (Figure 3b). Conversely, in HN01, GNAS mutations appeared as resistance developed. This dual observation underscores the importance of interpreting GNAS mutations within the broader context of tumor dynamics and patient-specific factors, rather than viewing them solely as markers of progression.

### 2.6. Detection of Emerging Amplifications in PD Patients and Clinical Implications

Two patients with PD, HN03 and HN06, displayed amplifications detected through ctDNA analysis at the second evaluation point under ICI therapy. In HN03, both EGFR and PIK3CA amplifications were identified, while in HN06, only EGFR amplification was observed. These amplifications were absent at baseline, and the appearance of gene amplification was consistent with tumor progression (Figure 3c). The detection of these amplifications exclusively through liquid biopsy, and not in tissue CGP, underscores the enhanced sensitivity of ctDNA in capturing real-time tumor evolution.

### 2.7. Comparative Analysis of Mutations Between Plasma and Tissue Samples

NGS analysis of tissue samples was performed for seven patients prior to subsequent treatments. We detected 14 copy number variants (CNVs) and 23 mutations in tissue samples, compared to 2 CNVs and 11 mutations in plasma (Table 1). 

The median number of detected genes was higher in tissue samples (4; range: 1–9) than in plasma (1; range: 0–4). Only 6.4% (3 of 47) of genetic abnormalities identified in tissue were also detected in plasma, and all matched cases were associated with PD.

## 3. Discussion

### 3.1. Clinical Utility of Liquid Biopsy in HNSCC

This study, as part of the SHIZUKU-HN initiative, assessed the clinical use of liquid biopsy for HNSCC based on data from the nationwide C-CAT genome database11. Notably, only 7% of cases utilized plasma specimens for comprehensive genomic profiling (CGP) compared to 10–20% in other cancers like pancreatic and biliary tract cancers, where tissue collection is particularly challenging. This discrepancy underscores a knowledge gap regarding the clinical utility of plasma ctDNA in HNSCC, particularly for monitoring treatment response. Currently, there are limited reports on ctDNA monitoring for drug therapy in HNSCC, both globally and in Japan, highlighting the importance of our study in exploring ctDNA’s potential as a tool for assessing ICI efficacy.

### 3.2. Correlation Between ctDNA VAF and Tumor Volume

This study showed the most significant correlation between mean VAF in pathogenic variants (pVAF) and tumor volume (Spearman’s ρ = 0.667, *p* = 0.001). This finding aligns with previous research in lung cancer [15,16], where mean VAF has been validated as a reliable marker of tumor burden and dynamics. Notably, changes in mean pVAF were detected early in the treatment course, often preceding radiographic progression, underscoring its potential as a real-time biomarker for assessing disease status and therapeutic efficacy. Among patients who achieved PR or SD, mean pVAF remained relatively stable, whereas in those with PD, early increases in mean pVAF accurately predicted disease progression. These results underscore the value of ctDNA monitoring for the early and accurate prediction of treatment outcomes. However, the variability in ctDNA detectability observed in this study highlights both the potential and limitations of ctDNA as a biomarker in HNSCC. In patient HN04, who had a small cervical lymph node lesion (<1 cm^3^), ctDNA was undetectable, suggesting reduced sensitivity in cases with minimal tumor burden. In contrast, patient HN09, despite having a larger lung lesion (31.3 cm^3^), also showed no detectable ctDNA, indicating potential site-specific limitations, particularly in lung metastases. Conversely, ctDNA was detectable in patient HN06, who had multiple lung metastases with a smaller tumor volume (2.14 cm^3^), suggesting that factors beyond tumor size, such as the number of metastatic lesions and tumor biology, may influence ctDNA shedding. Previous research suggested a complementary role of metabolic tumor burden and VAF in the evaluation of HNSCC [17]. These findings emphasize the need to integrate ctDNA analysis with other diagnostic modalities for comprehensive monitoring and treatment adaptation, especially in patients with low tumor volumes or lung metastases.

### 3.3. Dynamic Changes in Mutational Profiles: BRAF, APC, and GNAS

Our study revealed significant dynamic changes in BRAF, APC, and GNAS mutations during ICI treatment, shedding light on the tumor’s molecular evolution under therapeutic pressure. In patients with PR, both BRAF and APC mutations disappeared, suggesting these mutations were associated with subclonal populations that were particularly susceptible to ICI therapy. The loss of APC mutations, which are known to disrupt tumor-suppressive pathways and promote aberrant Wnt signaling, suggests a reduction in immune evasion mechanisms [18,19]. This shift could enhance treatment response by improving immune surveillance and T-cell activity. The sustained PR and continuous absence of APC mutations indicate a positive clonal shift, where less aggressive or more immunogenic tumor subclones dominate, maintaining efficacy over time. BRAF mutations, especially those that activate the MAPK pathway [20,21], can increase tumor immunogenicity and enhance T-cell infiltration, potentially sensitizing tumors to ICIs [22]. In HN01, the initial presence of a BRAF mutation coincided with a PR, suggesting that the mutation may have temporarily enhanced the tumor’s response to ICIs. However, as the disease progressed to PD, the BRAF mutation was no longer detectable, indicating selective pressure from treatment that led to clonal evolution. The disappearance of BRAF-mutated subclones might reflect adaptive resistance, where other resistant clones emerge, possibly involving pathways not captured in ctDNA [23].

The GNAS gene encodes a stimulatory G protein involved in the hormonal regulation of adenylate cyclase, and mutations frequently occur at codon 201, where arginine 201 is replaced by histidine (R201H) or cysteine (R201C). These mutations are associated with excessive proliferation and tumor development [24,25]. Wilson et al. demonstrated that the activating R201C mutation in GNAS promotes tumorigenesis in APC-inactivated mice via activation of the Wnt and ERK1/2 MAPK pathways [26]. This suggests that activating mutations in GNAS can cooperate with APC inactivation in driving tumorigenesis. The activation of GNAS may affect MAPK and Wnt signaling pathways, which are crucial in regulating cell growth, differentiation, and immune interactions. Although the detailed mechanisms of BRAF and APC loss and GNAS expression in this study are not fully understood, shifts in these important signaling pathways involved in tumorigenesis may contribute to changes in the tumor environment and could be associated with the ICI response. The activation of these pathways could potentially modulate the tumor microenvironment by promoting immune evasion, altering antigen presentation, or affecting cytokine signaling, thereby influencing treatment outcomes. Our observations of GNAS mutations in both responders and non-responders suggest a context-dependent role that highlights the complexity of tumor-immune interactions under ICI therapy. Future research should aim to clarify these mechanisms and explore targeted approaches to mitigate resistance associated with these genetic alterations.

### 3.4. Clinical Implications of Emerging Amplifications and the Role of Liquid Biopsy

The detection of EGFR and PIK3CA amplifications in patients HN03 and HN06 illustrates the dynamic nature of tumor evolution and the development of resistance under ICI therapy. These amplifications, identified exclusively through ctDNA and not tissue CGP, highlight the critical advantage of liquid biopsy in detecting emerging genetic alterations that drive disease progression. In HN03, the presence of both EGFR and PIK3CA amplifications suggests a complex resistance mechanism involving multiple pathways. EGFR amplification likely contributed to aggressive tumor growth, while PIK3CA amplification, through activation of the PI3K/AKT pathway, may have facilitated immune evasion and treatment resistance [27,28]. This combination underscores the tumor’s ability to adapt under therapeutic pressure and highlights the necessity of personalized monitoring to capture these changes. After the detection of EGFR amplification, switching to cetuximab resulted in significant tumor shrinkage, demonstrating the impact of real-time ctDNA findings in guiding effective treatment adjustment. In HN06, the identification of EGFR amplification without corresponding detection in tissue samples further emphasizes the utility of ctDNA in capturing key resistance mechanisms missed by traditional tissue biopsies. These findings highlight the potential of ctDNA to inform clinical decisions that directly impact patient outcomes by providing actionable insights into a tumor’s evolving genomic landscape. However, definitive conclusions cannot be drawn because Guardant360 does not differentiate between localized EGFR amplification and aneuploidy.

### 3.5. Concordance Between Plasma and Tissue Samples

The low concordance rate (6.4%) between ctDNA and tissue samples observed in our study is notably lower than the typically reported rates of 50–80% in other solid tumors such as lung and colorectal cancers [29,30], where spatial and temporal heterogeneity limits the overlap between plasma and tissue genomic profiles. Interestingly, concordance was primarily seen in PD cases, suggesting that ctDNA may better capture aggressive clonal populations driving disease progression. However, NGS tests used on plasma and tissue differ in terms of detectable genetic mutations, and more accurate comparisons should be made using the same test. On the other hand, the ctDNA analysis detected mutations not found in conserved tissues, such as EGFR amplification, which is useful information for the use of anti-EGFR therapies such as cetuximab. These findings reinforce the complementary role of ctDNA in capturing real-time genetic changes that guide treatment adjustments.

### 3.6. Study Limitations

Our study’s limitations include a small sample size (n = 10) and limited follow-up duration, both of which were constrained by study costs and the challenges of recruiting patients with recurrent/metastatic HNSCC. Additionally, while the cohort focuses specifically on oral cancer cases, variations in primary tumor size, treatment history, and metastatic profiles contribute to some degree of heterogeneity, which may introduce bias in interpreting the results. Nevertheless, compared to broader HNSCC trials that often encompass multiple primary sites, our study represents a relatively homogeneous subgroup analysis within HNSCC, emphasizing its exploratory nature. These limitations suggest that further validation in larger, multi-center studies is necessary. Technological improvements in ctDNA detection, particularly for patients with low tumor burden or challenging metastatic profiles, could enhance the sensitivity and clinical utility of ctDNA analysis in HNSCC. Addressing these limitations will be crucial for refining ctDNA’s role in clinical practice and optimizing its application for real-time treatment monitoring.

### 3.7. Clinical Implications and Future Directions

This study provides new evidence supporting the clinical utility of ctDNA monitoring in HNSCC, highlighting its potential as a non-invasive tool for real-time assessment of treatment response. The dynamic changes observed in key mutations such as BRAF, APC, GNAS, EGFR, and PIK3CA suggest that ctDNA analysis can offer unique insights into tumor evolution that are not captured by traditional tissue biopsies. These findings emphasize the role of ctDNA as a bridge between clinical practice and research, guiding personalized treatment strategies that adapt to the evolving genetic landscape of tumors.

Moreover, the observed correlation between ctDNA VAF and tumor volume reinforces the concept of ctDNA as a dynamic marker of tumor burden, potentially providing earlier indications of treatment efficacy compared to radiographic assessments. For example, pseudoprogression occurs when immunotherapy activates an immune response to the tumor, resulting in necrosis, hemorrhage, and edema [31,32], an event that can complicate the clinician’s judgment during tumor evaluation. The incidence of immunotherapy pseudoprogression in head and neck cancer is reported to be less than 2% [33], but it is indistinguishable from true tumor progression on imaging evaluation alone. In such cases, ctDNA analysis may provide a more accurate indication of treatment response than radiographic evaluation. This highlights the clinical importance of incorporating ctDNA monitoring into decision-making processes, particularly in settings where rapid adaptation of therapy is crucial. Despite current limitations, this study lays the groundwork for larger, multi-center trials that could further validate ctDNA as a reliable biomarker in HNSCC. As ctDNA detection technologies continue to advance, their integration into routine clinical workflows could enhance early intervention strategies, refine patient selection for targeted therapies, and ultimately improve clinical outcomes. The results suggest that ctDNA monitoring deserves broader clinical application and continued research to fully harness its potential as a transformative tool in oncology.

## 4. Materials and Methods

### 4.1. Analysis of C-CAT Database

To evaluate the clinical use of ctDNA testing, we reviewed the genomic information of cancer patients at the national core hospitals in the C-CAT database from 2019 to 2023. The C-CAT database11 included genomic data from the Foundation One CDx (F1CDx) test, F1LCDx test, and the National Cancer Center (NCC) Onco panel test. This study was approved by the C-CAT Information Use Review Committee (proposal control number: CDU2022-021N). We specifically extracted data relevant to ctDNA testing to assess its application across various cancer types, with a focus on head and neck squamous cell carcinoma.

### 4.2. SHIZUKU Study: Patient Recruitment and Monitoring

The SHIZUKU study (Systematic Hybrid Genomics Surveillance with Integrated Zonal Understanding and Kinetics Utility) is a prospective observational study conducted at our institution from October 2020 to December 2023. The study aims to evaluate dynamic mutation changes and variant allele frequency (VAF) in patients with HNSCC who are undergoing immune checkpoint inhibitor (ICI) therapy, with the primary endpoint being the correlation between changes in VAF and radiographic tumor response. The secondary endpoint focuses on identifying mutations associated with treatment resistance.

A total of 12 patients with recurrent or metastatic HNSCC who were scheduled to receive standard-of-care ICIs were enrolled; two patients were excluded due to unsuitable baseline samples, resulting in 10 patients being included in the analysis (Figure 4a, Table 2). The cohort included five males and five females, with a median age of 71 years (range: 52–92 years). Primary tumor sites included the tongue, gingiva, buccal mucosa, and other oral locations. Three patients received nivolumab as second-line therapy for platinum-refractory disease, and seven received pembrolizumab as first-line therapy. Tumor response was assessed using RECIST 1.1 criteria [34], with efficacy classified as complete response (CR), partial response (PR), stable disease (SD, lasting > 6 months), or progressive disease (PD). This study was approved by the institutional review board (approval number: G2020-020), and all participants provided written informed consent.

### 4.3. Plasma Collection and ctDNA Analysis

Plasma samples were collected at baseline, before ICI infusion, and at subsequent key time points (4 weeks, 6 months, and at progression) to capture early, mid, and late responses to therapy. Next-generation sequencing (NGS) analysis of samples was performed using Guardant360 (Figure 4b). Two 10 mL peripheral venous blood samples were collected in Streck Cell-Free DNA BCT (Streck, La Vista, Nebraska, USA) from each patient and sent to Guardant Health. All cfDNA extraction, processing, and sequencing were performed in a CLIA-certified and CAP-accredited laboratory. Guardant360 sequences 74 cancer-associated genes to identify somatic alterations. cfDNA was extracted from plasma, enriched for targeted regions, and sequenced using the Illumina platform with hg19 as the reference genome [35,36,37]. The detected genomic alterations included single nucleotide variants (SNV), gene amplifications, fusions, short insertions/deletions (up to 70 base pairs), and splice site-disrupting events [38,39].

We hypothesized that the VAF of ctDNA would correlate with body tumor volume. To test this, we analyzed the VAF values of pathogenic genes (pVAF) and the VAF values of all detected genetic variants (aVAF) including VUS and synonymous mutations. For each sample, somatic SNV from the Guardant360 report were used to evaluate the mean, maximum, and total VAF. Mean VAF was calculated by averaging the VAF for all variants detected in each blood draw.

### 4.4. Mutational Analysis of Tissue Specimen

Patients in this study underwent NGS analysis using tissue samples before transitioning to the next treatment. NGS analysis was conducted using the Foundation One^®^ Companion Diagnostic (F1CDx; Foundation Medicine, Inc., Cambridge, MA, USA) test [40]. The F1CDx test detects substitutions, insertions, deletions, and copy number alterations in 324 genes, as well as selective gene rearrangements and genomic signatures such as MSI and TMB. Samples utilized FFPE tumor tissue samples obtained from biopsies or surgery. In the case of biopsy, a portion of the tumor was excised, and a sufficient volume was collected for NGS testing. Pathologists selected suitable tumor specimens for testing. Tissue samples were collected before or after ICI treatment.

### 4.5. Radiology Total Body Tumor Volume Estimation

Tumor volume was calculated using the image analysis workstation (syngo.via) based on CT scans taken at approximately the same time as plasma sample collection (Figure 4c). Tumor areas were measured by manually setting the tumor contour in all cross-sections where the tumor was visible. The calculated tumor area was multiplied by the thickness of the image to obtain the volume. Total tumor volume was determined by summing the volumes of all sections.

### 4.6. Statistical Analysis

Statistical analyses were performed using the EZR (Easy R) software package version 1.54 (Saitama Medical Center, Jichi Medical University, Saitama, Japan) [41]. Spearman’s correlation coefficients were calculated to assess the relationship between ctDNA VAF and tumor volume. All statistical tests were conducted with a significance level of *p* < 0.05.

## 5. Conclusions

Despite its challenges, the SHIZUKU-HN study demonstrates the significant potential of plasma ctDNA as a real-time biomarker for monitoring ICI therapy in HNSCC. The strong correlation between mean VAF and tumor volume emphasizes ctDNA’s role in providing early insights into treatment efficacy, which could revolutionize personalized cancer management by informing therapeutic decisions ahead of traditional imaging methods.

## Figures and Tables

**Figure 1 ijms-26-00235-f001:**
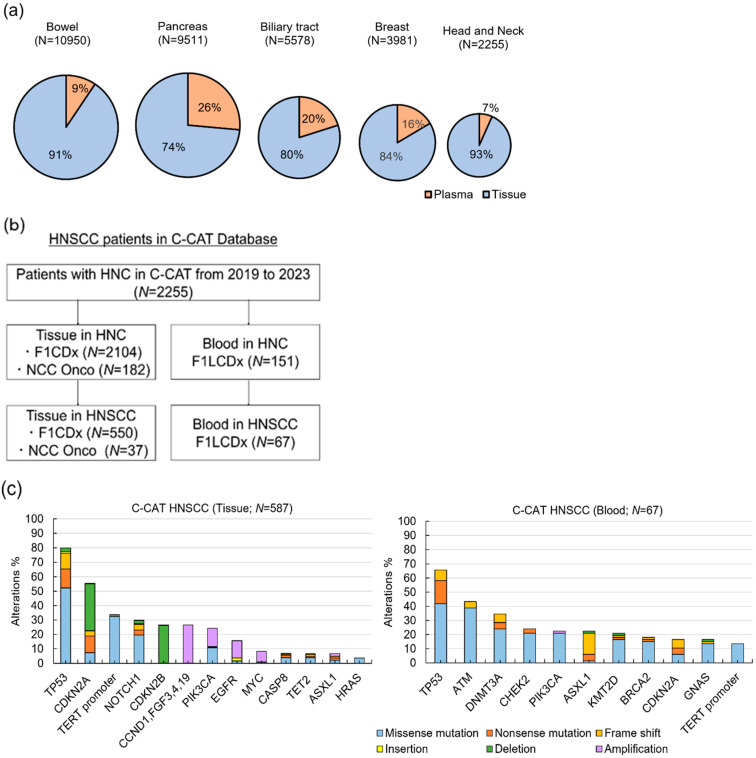
(**a**) Distribution of panel tests based on tissue (blue) and blood (red) samples in the C-CAT database. (**b**) Representation of the head and neck squamous cell carcinoma (HNSCC) cohort within the C-CAT database. (**c**) Frequently detected gene variants in tissue and blood samples from HNSCC cases. CGP, comprehensive genomic profiling; C-CAT, Cancer Center for Advanced Cancer Genome Therapy; HNC, head and neck cancer; HNSCC, head and neck squamous cell carcinoma.

**Figure 2 ijms-26-00235-f002:**
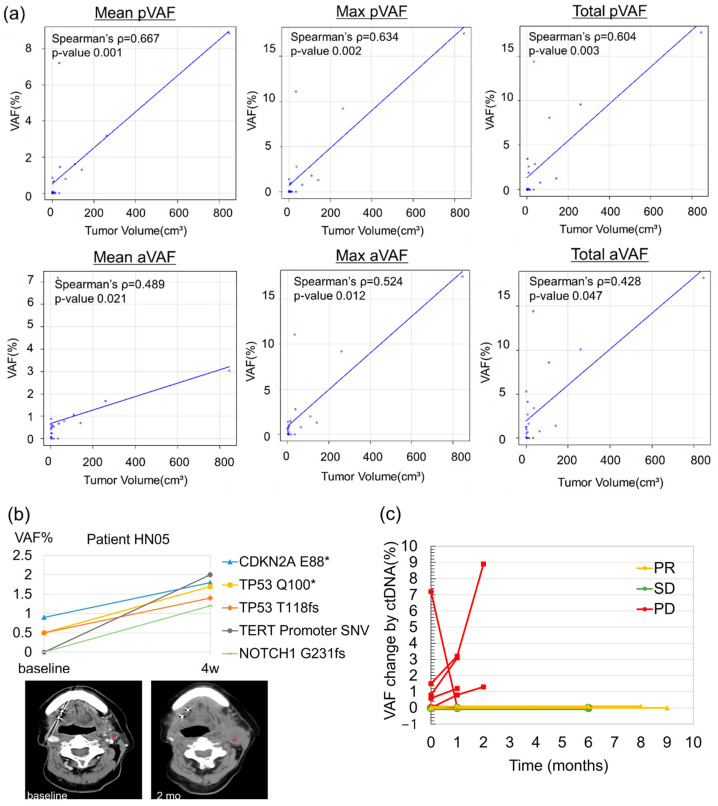
Correlation Between ctDNA and tumor volume, and comparison with tissue-based analysis. (**a**) Correlation between tumor volume (cm^3^), as determined by CT volumetric analysis, and mean VAF across 27 serial plasma samples from 10 patients. Pathogenic VAF (pVAF) represents VAF for pathogenic mutations, while all VAF (aVAF) includes all somatic mutations, encompassing variants of unknown significance (VUS) and synonymous mutations. (**b**) CtDNA dynamics for each gene variant, alongside the corresponding radiographic response, illustrated for a progressive disease case (HN05). Asterisks on CT images indicate target lesions. (**c**) Spider plot depicting mean VAF dynamics during ICI treatment, with color-coded radiographic responses: PD (red), SD (green), and PR (yellow). VAF, variant allele frequency; VUS, variants of unknown significance.

**Figure 3 ijms-26-00235-f003:**
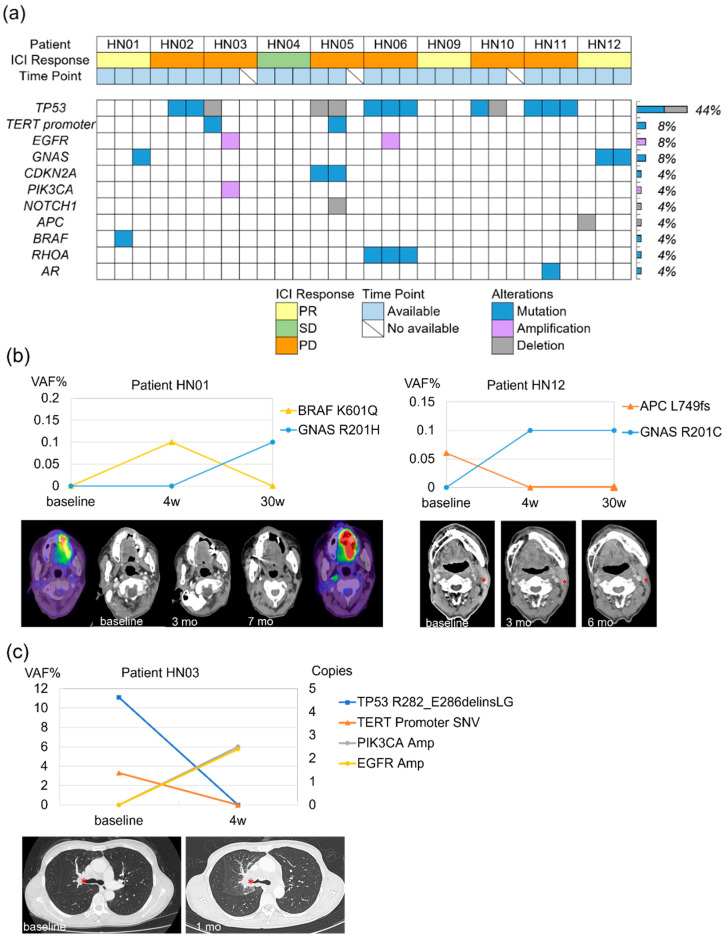
Study design and sequencing outcomes at our institution. (**a**) Detected plasma variants through next-generation sequencing (NGS), color-coded by ICI response, sample collection point, and type of genetic alteration. (**b**,**c**) Dynamic ctDNA patterns for each gene variant and the corresponding radiographic response, highlighting HN01 and HN12, which demonstrated partial response (PR). HN03 showed progressive disease (PD). Asterisks on CT images indicate target lesions. HN01 is shown as a PET/CT image to clarify tumor lesions. R/M HNSCC, recurrent/metastatic head and neck squamous cell carcinoma; NGS, next-generation sequencing.

**Figure 4 ijms-26-00235-f004:**
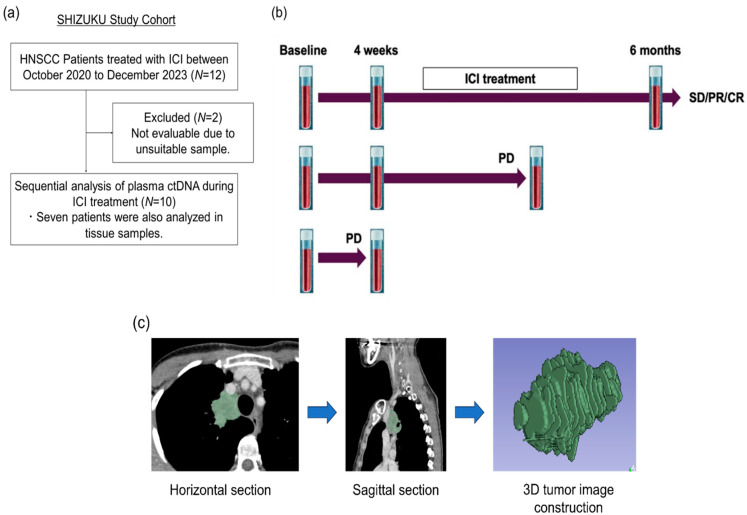
Timing of plasma collection and body tumor volume estimation. (**a**) Patients monitored by ctDNA analysis at our institution; 12 patients with recurrent/metastatic head and neck squamous cell carcinoma (R/M HNSCC) who received immune checkpoint inhibitors (ICIs) were enrolled; 2 patients were excluded due to inadequate baseline samples. (**b**) Plasma ctDNA analysis using Guardant360. Plasma samples were obtained before ICI infusion (baseline), at 4 weeks, 6 months, or at progression of disease (PD). (**c**) Tumor volume measurement using computed tomography (CT) volume analysis on an image analysis workstation (syngo.via). Tumor contours were set in the axial and sagittal planes as shown. ctDNA, circulating tumor DNA; ICIs, immune checkpoint inhibitors; CT, computed tomography.

**Table 1 ijms-26-00235-t001:** Comparison of tissue and plasma.

Case No.	Plasma *	Tissue	Best Response
HN01	GNAS R201H	TP53 R248Q NOTCH1 E1305* CDKN2A splice site SNV	PR
HN02	TP53 splice site SNV	TP53 splice site SNV CDKN2A/B loss	PD
HN03	TP53 R282_E286>LG TERT Promoter SNV PIK3CA amp EGFR amp	TP53 R282_E286>LG	PD
HN04	(VUS only)	TP53 I232fs*8 TP53 C242S CDKN2A/B loss CCND1, FGF3,4,19 amp EPHB4 amp GATA6 amp	SD
HN06	TP53 I251T TP53 R213* TP53 C242fs TP53 H179Q RHOA E40Q	TP53 I251T TERT promoter-124C>T NOTCH1 C1007* NOTCH3 T56fs*11 CDKN2A V28_E33del CCND1, FGF3,4,19 amp	PD
HN09	None	TP53 S241C CCND1, FGF3,4,19 amp IRF2 R110>P* TBX3 S55*	PR
HN12	APC L749fs GNAS R201C	CDKN2A p16INK4a P114L TERT promoter-124C>T TET2 L1340R TP53 N311fs*34	PR

Underline indicates matched mutations. * Genetic mutations detected in plasma during ICI treatment are shown all together.

**Table 2 ijms-26-00235-t002:** Clinical characteristics.

Characteristic		Number (%)
All patients		10
Age		
	Median (range, y)	71 (52–92)
Gender		
	Male	5 (50)
	Female	5 (50)
Primary site		
	tongue	3 (30)
	gingiva	3 (30)
	buccal mucosa	1 (10)
	oral multiple primary	3 (30)
ECOG performance status		
	0	5 (50)
	1	5 (50)
Smoking status		
	Never or <10 packs/year	4 (40)
	Former (≥10 packs/year)	6 (60)
Cancer staging		
	Stage III	1 (10)
	Stage IV	9 (90)
HPV		
	Negative	10 (100)
	Positive	0
ICI drug		
	Nivolumab	3 (30)
	Pembrolizumab	7 (70)

## Data Availability

The original contributions presented in this study are included in the article. Further inquiries can be directed to the corresponding author(s).

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
