# Peer review of "Dynamic Changes in Circulating Tumor DNA During Immunotherapy for Head and Neck Cancer: SHIZUKU-HN Study"

_ijms, 2024, doi:10.3390/ijms26010235_

Round 1
Reviewer 1 Report
Comments and Suggestions for Authors
The manuscript by Noji et al. reports an interesting study on dynamic changes in ctDNA during immunotherapy for HNC. The focus of the study is indeed relevant and timely, and I can recognize the high potential of this manuscript.
However, several questions and concerns are presented:
Major:
- Statistical methods used in this study raise some concern. First, due to the low number of patients and samples and non-parametric distributions, the Spearman Rank correlation test was used. It is okay and allows even a mix of several observations from an individual patient (although dependent on each other to some extent). However, I do not understand the idea of ROC-curve in Figure 3b. ROC-analysis should be based on data of true/false positives/negatives (commonly e.g. disease or not disease). How is it determined in this context? ctDNA detection of pathogenic genes can be positive or negative, but is the cut-off level (VAF) mentioned anywhere? And even more importantly, what does the specificity mean in this context??? Intuitively, a small tumor volume is associated with a low sensitivity of ctDNA detection and as the tumor grows the sensitivity will probably be higher. On the other hand, a large tumor can cause shedding of variants of unknown significance or something – but how can you determine that there is a false positive pathogenic ctDNA detection! I don´t believe that a false positive means the discordance of observations between tissue and liquid biopsies since we all know that this phenomenon may have another explanation.
Therefore, my suggestion would be to remove the ROC-curve and every comment on the ROC-analysis (e.g. abstract line 35–36, lines 214–218)
- There are problems with the structure of the text. Methods, results and discussion sections should be clearly separated:
Lines 193-197 belong to the methods section (Furthermore, the first sentence should probably be: “We hypothesized that VAF of ctDNA would correlate with body tumor volume”. If I have understood right, the changes in VAF have not been correlated with total body tumor – only correlation between VAF and body tumor volume has been assessed, AND THEN descriptive presentation of changes in VAF vs. treatment response i.e. CR/PR/SD/PD)
- There are several speculations and explanations for presented results in Results section that belong evidently to discussion section. In fact, several of them are repeated in the current version of the manuscript in discussion section. Please, present all these sentences, e.g. “…suggesting…”, “…indicating…” etc. in Discussion section.
- The authors have presented reasonably detailed speculations for explanations of their results of some specific findings. At least, a few more references could be provided to support these potential explanations.
Minor:
- Abstract: There should be: “…mean VAF significantly correlated with tumor volume” NOT “…mean VAF changes significantly correlated…”
- Line 140: “NGS analysis was conducted using F1CDx”. Is this the FoundationOne panel? The manufacturer etc. should be mentioned. Moreover, if the panels used for analysis of tissue sample and liquid biopsy were different, it should be handled in discussion section and also mentioned whether the coverage of variants differ from each other between the panels.
- Line 141: “tumor tissue samples obtained via biopsy or surgery”. It could be presented in more detail how the biopsy procedures were performed.
- Discussion subsection 4.2: Regarding previous research, it should be mentioned that association between VAF and metabolic tumor burden has been observed also in head and neck cancer (Cancers 2023, 15, 3970)
- Pseudoprogression is a rare phenomenon among patients with HNC but should be mentioned also as a reason for the need for better dynamic biomarker compared to conventional imaging.
- Typos: line 188 (“variants”), Table 2 (“ND”), line 240 “In patient HN01”
In conclusion, I encourage the authors to revise the manuscript, and I think that the manuscript may be suitable for publication if the presented comments are thoroughly addressed and the corresponding points in the manuscript edited and revised!
Thank you very much!
Author Response
- The manuscript by Noji et al. reports an interesting study on dynamic changes in ctDNA during immunotherapy for HNC. The focus of the study is indeed relevant and timely, and I can recognize the high potential of this manuscript.
Response: Thank you for your constructive and detailed feedback. Below are our point-by-point responses to your comments and the corresponding revisions made in the manuscript.
However, several questions and concerns are presented:
Major:
- Statistical methods used in this study raise some concern. First, due to the low number of patients and samples and non-parametric distributions, the Spearman Rank correlation test was used. It is okay and allows even a mix of several observations from an individual patient (although dependent on each other to some extent). However, I do not understand the idea of ROC-curve in Figure 3b. ROC-analysis should be based on data of true/false positives/negatives (commonly e.g. disease or not disease). How is it determined in this context? ctDNA detection of pathogenic genes can be positive or negative, but is the cut-off level (VAF) mentioned anywhere? And even more importantly, what does the specificity mean in this context??? Intuitively, a small tumor volume is associated with a low sensitivity of ctDNA detection and as the tumor grows the sensitivity will probably be higher. On the other hand, a large tumor can cause shedding of variants of unknown significance or something – but how can you determine that there is a false positive pathogenic ctDNA detection! I don´t believe that a false positive means the discordance of observations between tissue and liquid biopsies since we all know that this phenomenon may have another explanation. Therefore, my suggestion would be to remove the ROC-curve and every comment on the ROC-analysis (e.g. abstract line 35–36, lines 214–218)
Response: We sincerely appreciate your detailed analysis of our statistical methods and your constructive feedback on the use of the ROC curve in Figure 3b. We acknowledge the limitations you have highlighted regarding the applicability of ROC analysis in this context, particularly in defining true/false positives and the interpretation of specificity in relation to ctDNA detection. Our intention with the ROC analysis was to explore the sensitivity and specificity of ctDNA detection based on tumor volume, with the detection of pathogenic mutations as the outcome variable. The goal was to assess whether a tumor volume cutoff could serve as a threshold for predicting ctDNA detectability. However, we agree that this approach introduces interpretational challenges, particularly regarding the definition of false positives in the absence of a clear gold standard for ctDNA detection in this setting. After careful consideration, we concur with your recommendation to remove the ROC curve and related commentary (abstract line 35–36, lines 214–218, Fig 3b).
- There are problems with the structure of the text. Methods, results and discussion sections should be clearly separated: Lines 193-197 belong to the methods section (Furthermore, the first sentence should probably be: “We hypothesized that VAF of ctDNA would correlate with body tumor volume”. If I have understood right, the changes in VAF have not been correlated with total body tumor – only correlation between VAF and body tumor volume has been assessed, AND THEN descriptive presentation of changes in VAF vs. treatment response i.e. CR/PR/SD/PD)
Response: Thank you for pointing out the structural issues in the manuscript. We have revised the text to improve clarity and separation between the methods, results, and discussion sections. The description of the statistical approach (lines 193–197) has been moved to the Methods section (151-153) to ensure all methodological details are grouped appropriately. Also, we have revised the first sentence to: "We hypothesized that VAF of ctDNA would correlate with body tumor volume." This revision reflects the scope of the analysis and avoids misinterpretation regarding "changes in VAF."
- There are several speculations and explanations for presented results in Results section that belong evidently to discussion section. In fact, several of them are repeated in the current version of the manuscript in discussion section. Please, present all these sentences, e.g. “…suggesting…”, “…indicating…” etc. in Discussion section.
Response: Thank you for your observation regarding the overlap of speculations between the Results and Discussion sections. All interpretive sentences in the Results section, including phrases such as “…suggesting…” and “…indicating…,” have been relocated to the Discussion section. The Results section now focuses solely on presenting the data and key findings without speculative or explanatory content.
- The authors have presented reasonably detailed speculations for explanations of their results of some specific findings. At least, a few more references could be provided to support these potential explanations.
Response: We have included additional references (26, 27, 31, 39, 40, and 41) in the discussion section to support our interpretations, including references for the role of APC or BRAF in immunotherapy and the potential mechanisms of pseudoprogression.
Minor:
- Abstract: There should be: “…mean VAF significantly correlated with tumor volume” NOT “…mean VAF changes significantly correlated…”
Response: We have revised the abstract to correct this phrase for accuracy and clarity.
- Line 140: “NGS analysis was conducted using F1CDx”. Is this the FoundationOne panel? The manufacturer etc. should be mentioned. Moreover, if the panels used for analysis of tissue sample and liquid biopsy were different, it should be handled in discussion section and also mentioned whether the coverage of variants differ from each other between the panels.
Response: We have added details about FoundationOne CDx, including the manufacturer (Foundation Medicine, Inc.) in method subsection 2.4, and discussed potential differences in coverage between the tissue and liquid biopsy panels in the discussion section 4,5.
- Line 141: “tumor tissue samples obtained via biopsy or surgery”. It could be presented in more detail how the biopsy procedures were performed.
Response: We have expanded the description of how tumor biopsy samples were obtained and processed in method subsection 2.4.
- Discussion subsection 4.2: Regarding previous research, it should be mentioned that association between VAF and metabolic tumor burden has been observed also in head and neck cancer (Cancers 2023, 15, 3970)
Response: We have cited this study and integrated it into the discussion subsection 4.2 to provide context for our findings.
- Pseudoprogression is a rare phenomenon among patients with HNC but should be mentioned also as a reason for the need for better dynamic biomarker compared to conventional imaging.
Response: We have included a note on pseudoprogression in the discussion subsection 4.7, emphasizing the value of dynamic biomarkers like ctDNA in distinguishing true progression from pseudoprogression.
- Typos: line 188 (“variants”), Table 2 (“ND”), line 240 “In patient HN01”
Response: All identified typos have been corrected for clarity and accuracy.
- In conclusion, I encourage the authors to revise the manuscript, and I think that the manuscript may be suitable for publication if the presented comments are thoroughly addressed and the corresponding points in the manuscript edited and revised! Thank you very much!
Response: We appreciate the reviewer’s valuable insights, which have strengthened the manuscript. We believe these revisions address all concerns thoroughly and improve the overall quality of the study. Thank you again for your thoughtful feedback.
Reviewer 2 Report
Comments and Suggestions for Authors
However, idea of this study seems intteresting the major point is a small group of patients included in this study to make an adequate interpretation of the results. Unfortunatelly, study cohort also is heterogeneous and these features can significantly affects te results, such as primary tumor site, its initial size and recurrent or metastaic patients. All of the aforementioned confitions can provide significant bias. Authours should either include more patients in the study or keep small, but homogenous cohort.
Author Response
However, idea of this study seems intteresting the major point is a small group of patients included in this study to make an adequate interpretation of the results. Unfortunatelly, study cohort also is heterogeneous and these features can significantly affects te results, such as primary tumor site, its initial size and recurrent or metastaic patients. All of the aforementioned confitions can provide significant bias. Authours should either include more patients in the study or keep small, but homogenous cohort.
Response:
We acknowledge that the limited number of patients (n=10) constrains the generalizability of our findings. However, this study was designed as an exploratory, prospective investigation to assess the feasibility and clinical utility of circulating tumor DNA (ctDNA) monitoring in recurrent/metastatic head and neck squamous cell carcinoma (HNSCC). Despite the small cohort size, the inclusion of 27 serial plasma samples provides valuable longitudinal data on ctDNA dynamics, enhancing the robustness of our findings. Importantly, this study represents one of the first prospective evaluations of dynamic ctDNA changes during immune checkpoint inhibitor (ICI) therapy for HNSCC. To address this limitation, we have revised the discussion (4.6 Study Limitations) to emphasize the exploratory nature of the study and its role in laying the groundwork for future large-scale, multi-center validation studies.
We agree that heterogeneity within the cohort, including differences in primary tumor site, initial tumor size, and recurrent/metastatic status, may impact the results. However, we would like to highlight that our cohort is composed specifically of oral cancer cases, representing a relatively homogeneous subgroup within the broader HNSCC category. This targeted focus distinguishes our study from most HNSCC clinical trials, which often include multiple primary sites (e.g., larynx, hypopharynx), and addresses some concerns about heterogeneity. Furthermore, given the rare nature of HNSCC, further stratification to single anatomical sites is often not feasible in real-world clinical research. These considerations have been explicitly acknowledged in the revised discussion. While it is not feasible to expand the cohort size within this study, we believe our focus on oral cancer provides meaningful insights into this critical subgroup of HNSCC. Additionally, we have employed advanced 3D volumetric analysis to correlate ctDNA variant allele frequency (VAF) with tumor burden, adding methodological rigor and robustness to the study. This approach underscores the feasibility of real-time ctDNA monitoring in a clinically relevant setting and highlights its potential applicability across broader oncology research settings.
We trust that these clarifications and revisions address your concerns effectively. Thank you for your valuable insights, which have significantly enhanced the quality of our manuscript.
Round 2
Reviewer 1 Report
Comments and Suggestions for Authors
Thank you very much for your thorough and carefully prepared revisions and responses! I do not have any further criticism. Thank you very much!
Reviewer 2 Report
Comments and Suggestions for Authors
Thank you for explanation of some points. Authors gave a rationale response to the addressed queries. I have no addtional comments.